# Gathering Stories: Creating Spaces for Young Women to Connect and Build Community through Multimodal Storytelling

Laura Shackelford [1,*], Ammina Kothari [2,*] and Karen vanMeenen [1,*]

1   Department of English, Rochester Institute of Technology, Rochester, NY 14623, USA
2   Department of Journalism, University of Rhode Island, Kingston, RI 02881, USA
*   Correspondence: lxsgla@rit.edu (L.S.); akothari@uri.edu (A.K.); kmvgla@rit.edu (K.v.)

**Abstract:** Digital storytelling prioritizes real-time connections, story creation, contextual adaptability, multi-media expression, and accessibility. This article discusses the unrecognized affordances and value of digital storytelling practices for teens living in precarious (neo)colonial lifeworlds. We review the workshop methods developed as designers and leaders of "Gathering Stories: A Digital Storytelling Workshop for Young Women" in July 2021 to enliven and illuminate high school students' voices while also addressing social, emotional, and affective experiences and needs during the pandemic. The article details how we co-realized spaces where teens' lived experience of gathering and the draw of story were the driving forces for their diverse storytelling practices. Identifying positive outcomes for the first iteration of the workshop, we also identify challenges that will inform future iterations of the workshop, such as structural dimensions of intersectionality and the challenges predicative AI such as ChatGPT poses to such efforts to prioritize experiential dimensions of learning through storytelling.

**Keywords:** digital storytelling; teen social and emotional development; high school writing; women's writing

## 1. Introduction

Flickering across ephemeral screens, digital stories appear to relinquish their status as cultural heritage 'objects' that sustain and empower via their longevity and stability, especially for young adults. Digital storytelling, instead, prioritizes real-time connections and story creation, contextual adaptability, multi-media expression, and accessibility. With a smartphone and Internet access, even young creators, with a bit of practice, can instantly post a story, create or remix a video, or capture and narrate an event in real time. Thanks to GPS-location technologies and mobile cellular phones, storytelling can happen nearly anywhere, which brings storytelling into proximity with lived experience in newly intimate, entangled ways. How might we, as educators, draw on the dynamic, multimodal affordances of digital storytelling as a way to engage high school students and enable them to recreate personally and culturally meaningful connections, building a story-based sustenance for the twenty-first century out of what might seem to be its ruins? How might we draw on teens' digital lives to expand their fluency with creative story practices in multiple genres, enabling them to build sustaining connections with other local teen storytellers while counteracting their sense that nobody's listening to them? How might the multimodality of digital storytelling speak to and catalyze more diverse creative expression, writing, cultural contexts, and cognitive styles—counteracting a white Euro-American, colonialist fetishization of abstract, transcendent writing (and its distrust of the visual and aural)?

In early 2021, in the thick of COVID-19 pandemic lockdowns and restrictions, as faculty members directing and affiliated with the Center for Engaged Storycraft at the Rochester

Institute of Technology (RIT), we recognized digital storytelling—and the shift of storytelling to networked and real spaces where even new creators can create multimedia digital stories—as a profound opportunity to engage high school women through storytelling. Research since then has thoroughly documented the overwhelming impact the pandemic has had on youth and on teenage women in particular (Barry 2023), though it is early days for a definitive assessment of the complex forces—both pre- and post-pandemic—that are negatively affecting so many teen women. In our own community of Rochester, New York, it was apparent that the sudden transition to virtual learning platforms during the height of the pandemic resulted in the loss of social and emotional support outside of teens' immediate family members. We thus turned to multimodal storytelling as a way to facilitate engagement among high schoolers with their peers and to identify common ground from which they could share experiences, however diverse.

Responding to a call from the Cornelia T. Bailey Foundation's New ERA Women Writers Grant, we developed and, with their funding, launched the first "Gathering Stories: A Digital Storytelling Workshop Experience for Young Women" in July 2021 on the RIT campus. We focused on high school women with an interest in storytelling and invited them to apply to participate in the week-long workshop. Our goal was to offer training in digital storytelling of diverse kinds and to facilitate a gathering of youth identifying as women—including transgender, nonbinary, and cisgender women—over the course of a week. The workshop provided the physical and interpersonal space, time, and resources for participants to explore, articulate, and connect their voices and emotional experience to a community of local women—including peers and elders—so that they would feel recognized and valued as storytellers and each could learn how to prioritize their own experience and voice as a leading, transformative driver in their educational, personal, and professional lives. In this paper, we discuss our approach to developing this digital storytelling workshop for young women, describe positive outcomes, highlight challenges we experienced in creating an equitable and collaborative storytelling environment, and finally, detail our plans for future iterations of the program.

## 2. Theoretical Framework

Much contemporary interest in storytelling and the approaches to storytelling that inform our workshop emerge out of late twentieth-century narrative studies by French (post)structuralist linguists, literary scholars, and philosophers such as Roland Barthes and Jean-François Lyotard, who turned attention to the narratives that inform everyday life in addition to the more artful, crafted narratives we have long celebrated as literature, folktales, or myth (Lyotard 1984; Barthes 1972). Barthes's influential 1966 "Introduction to the Structural Analysis of Narratives" recast narrative and, particularly, narrative in its everyday practice to include narrative discourses we carry out with ourselves and the world around us (Barthes and Duisit 1975). Formal or informal, artful or bland, narrative, from this vantage point, can unfold through "an almost infinite diversity of forms"; narratives are "able to be carried by articulated language, spoken or written, fixed or moving images, gestures, and the ordered mixture of all these substances." Narrative, in other words, is multimodal, drawing on more than one semiotic and sensory channel. As Barthes initiates a new discipline of narratology that makes us all (un)knowing storytellers, he also expands the realm of narrative to include diverse media, genres, and creators and identifies narrative "in every age, in every place, in every society"; famously stating that narrative is "simply there, like life itself" (p. 237).

In this vein, our workshop aims to empower the individual storyteller within each high school participant. The primary emphasis in the storytelling practice and final project is to empower each young woman to reflect on their personal experience, experiment with narrative media and methods to express themselves, and communicate something meaningful to an audience of peers. The value is largely in the interpersonal process of writing about oneself, experimenting with diverse narrative forms to compellingly craft and shape one's story, and personally sharing it with a group of peer and elder storytellers—not

to create a great work of literature, though this is often a longer-term goal for many of these young storytellers. As its title, "Gathering Stories," suggests, our workshop identifies and prioritizes storytelling and listening as a central means of connecting with others through shared experience, a means of personally and collectively gathering together the pieces of our existence for shared enjoyment and reflective benefit.

Barthes's and other narratologists' attention to the universality of narrative (across media and cultures) and their recognition of its centrality to human cognition, unfolding "like life itself" in our minds, led literary scholars such as Peter Brooks to explore how closely our sense of self is "bound up with the stories we tell about our own lives and the world in which we live" (Brooks 1996, p. 19). "Narrative capability shows up in infants some time in their third or fourth year, when they start putting verbs together with nouns... a conjunction that has led some to propose that memory itself is dependent on the capacity for narrative" (Abbott 2021, p. 3). Interest in narrative as a primary means through which we try to understand the world and construct its meanings quickly spread to psychology and then to other fields including medicine and economics, a movement since described as a "narrative turn" in the social sciences, instigated by Jerome Bruner's study of child development through the lens of children's informal storytelling among themselves and with adults in "The Narrative Construction of Reality" (Bruner 1991). Joseph Campbell similarly identifies an oft-repeated narrative structure—the "hero's journey"—across cultures and time, noting its direct parallels to stages of personal development and maturation, providing a well-known example of research on narrative's dynamic influence on personal development (Campbell 1973).

More recent interdisciplinary story work emerging from this "narrative turn" remains attentive to storytelling as a way people actively (re)conceptualize their life experiences; it recognizes the centrality of personal narratives to identity formation and, therefore, often mobilizes story as a key to behavior change, both individual and societal (see Shaffer et al. (2018) for a recent model for implementing and assessing the influence of narrative in behavioral medicine). As cognitive narratologists suggest, storytelling is often *enactive*—i.e., stories serve as intersubjective thinking/acting tools that connect tellers and listeners and, in doing so, provide models and potential trajectories of action, connecting past, present, and future states (Herman 2013, pp. 48–49).

Inviting high school women to participate in the workshop and reflect on their own stories in the year(s) preceding their transition from high school to college or professional life, we are attuned to narrative as a "*principal way in which our species organizes its understanding of time*" (Abbott 2021, p. 3) and, therefore, as an especially useful way for these young women to explore and build narrative bridges between their own past, present, and future, the three-part structure of a narrative's dramatic arc—its beginning, middle, and end, as first identified by Aristotle. We underscore the emotional shape of a narrative's rising and falling story arc as it moves through time and space like a smooth arc or a more dramatic, winding rollercoaster, encouraging these young women to envision the dramatic, emotional "shape of their story" (Vonnegut 2004) of their, at once, personal and narrative transformation. Our workshop participants are given complete freedom in their choice of a story to tell, as long as it draws from their experiences, and yet over ninety percent of the final stories, regardless of their diverse media, feature a pivotal moment of personal realization or maturation, or prepare for it by reflecting on a childhood past, a present, and an as-yet-unknown near-future. While this illustrates a familiarity with personal storytelling conventions, the teen storytellers' emotional investment in the particular stories they choose to tell also evidences their awareness of narrative's power to meaningfully shape their experiences and, as a result, provide a reflexive means of revisioning themselves and their relationship to a broader community.

Importantly, there are continuing, necessary debates about the extent to which narrative informs all people's sense of self, even if it seems neurotypical (Strawson 2018). In addition, the narrative turn and narrative constructivism that grew out of late twentieth-century narratology and social sciences have been rightly critiqued for taking the universality of

narrative so far that it eclipses non-narrative modes of communication (such as the lyric) altogether or entraps us in a world that is exclusively narrative, with no access to material reality outside the text (Shackelford 2014). Peter Brooks's *Seduced By Story: The Use and Abuse of Narrative* (Brooks 2022) describes his disabused sense of what has happened to narrative in the last century, describing a harrowing "narrative takeover of reality" in the twenty-first century as stories abound: "[n]arrative seems to have become accepted as the only form of knowledge and speech that regulates human affairs" (pp. 4, 3). While Brooks is correct in his assessment of how and why stories abound in such destructive ways today, particularly in political and economic domains, his call for a more nuanced and "intelligent account of what narrative is and does" (p. 4) is precisely the route already taken by twenty-first-century research in cognitive and in digital narratology, which continue to explore the transformative power of narrative, while providing important correctives to the narrative turn and its constructivist extremes by acknowledging the material environments, media, and events that reciprocally co-inform and co-realize narrative practices of self-understanding and of knowledge-building in given contexts or by distinct selves and storytellers. In *Storytelling and the Sciences of the Mind,* cognitive narratologist David Herman proposes two verbs as terms to capture the interweaving of narrative worlds and actual worlds (Herman 2013). "Storying the world" describes the active, reciprocal processes through which we use our narrative frameworks, experience, and assumptions as a kind of extended-mind to make sense of events in our material lifeworlds (p. 225). Alternatively, "worlding the story" involves processes through which we actively interpret narratives by asking empirical questions about characters and events in stories, imagining who, what or why things unfold as they do (p. 100). In both instances, narrative understanding relies on an exchange between narrative storyworlds and lifeworlds rather than granting narrative full or unlimited rein over material actualities.

Situating our young storytellers in the same physical workshop space daily over the course of a week and inviting local storytellers from the Rochester, New York, area, the workshop attempts to ground our storytelling practices in these specific contexts and to acknowledge the way geographical location, physical space, interpersonal dynamics, writing tools, and workshop leaders' guidance all inform our storytelling practices, as individuals and as a group. Researchers in Spain, recording human brain activity during verbal communication and storytelling sessions, recently discovered that the production/comprehension coupling they observed "resembles the action/perception coupling observed within mirror neurons" and thus a speaker's activity "is spatially and temporally coupled with the listener's activity" (Stephens et al. 2010). Through quantitative measuring of story comprehension, they concluded that "the greater the anticipatory speaker–listener coupling, the greater the understanding"—of story. Sharing stories deeply and *physiologically* connects individuals, something we cultivate over the course of our workshop through a series of small-group story-sharing activities.

In addition, guest storytellers model their brainstorming and the writing processes they use to draw on personal experience and identity and craft a meaningful, shareable story. These elder storytellers address the vulnerability and reflect on the questions that arise when transferring one's experience to a narrative, shareable form. "Storying the world" is an active process with choices of many kinds to be made, not a simple act of recording a pre-existing story or narrative verbatim. And, notably, it often requires our storytellers to break out of existing frames and conventions to express their very disdain for self-narrative or the narrativization of life.

It is here that the workshop's focus on digital storytelling and feminism and gender studies is particularly pertinent. Digital storytelling—or storytelling that relies on the application of digital software and communication technologies, such as the Internet, for its creation as well as its dissemination and reading—as an emerging, multimodal medium for storytelling, requires creators and scholars to attend closely to the material affordances *and* constraints that are introduced in these new "writing spaces," to borrow J. David Bolter's early term for digital writing environments (Bolter 2001). In contrast to print conventions

that, over time, seemed to be absolute, transcendent necessities of storytelling, digital storytelling has challenged and upended many of these conventions, like turning pages by moving from left to right as opposed to scrolling down, including only typed language in black font on a white page, or having the text remain stable over time. While digital writing environments, digital cultures, and cyberspace more generally were initially presumed to be liberatory and democratizing to their core, digital storytelling scholars since Janet H. Murray's *Hamlet on the Holodeck*: *The Future of Narrative in Cyberspace* (Murray 1998), and, more recently, the work of Marie-Laure Ryan and collaborators in her edited collection *Narrative across Media: The Languages of Storytelling* (Ryan 2004) take advantage of the newness of digital storytelling media as the basis for a comparative understanding of these storytelling technologies as their media-specific expressive capabilities and limits both enable and constrain writing as a result of their material and technological infrastructure and software. For instance, Ryan's typology of media affecting narrativity underscores the spatio-temporal dimensions of digital narratives and their inclusion of multiple sensory channels of communication at once as differentiating features of these narrative media, as in web comics, visual narratives, or interactive narratives, for example.

Whereas storytelling in print formats and cultures privileges text and a single, linear narrative movement through time, the workshop story prompts, final project, and guest storytellers engage multimodality to unsettle and question the hierarchical privileging of symbolic language and its alignment with a single, universal, transcendent truth within Euro-American colonialism, which has delegitimized and denigrated acoustic and visual modes of expression and the cultures that revere them. The symbolic register of language and the written word are also gendered masculine and racialized as white within Euro-American colonialist contexts, which initially led cyberfeminist and digital narrative scholars to believe that the introduction of non-linear, multimodal, interactive digital writing environments might equalize the previously gendered roles of the (active, masculine) author and the (passive, feminine) reader. Instead, this heteronormative, cis-gendered opposition is increasingly displaced by more diverse subject positions and identities, and all of these seem further differentiated and delimited, not enabled or empowered by the software and social media platforms on which they rely for their self-writing and storytelling. For this reason, our workshop draws on feminist theories of writing, such as Sara Ahmed's outlined in "Orientations Matter," which understand writing as a resoundingly material and symbolic practice that historically and contemporaneously relies on putting writing technologies and tools designed for others' benefit and uses to your own uses in the best traditions of women's writing from Virginia Woolf's 1929 essay "A Room of One's Own" to Barbara Smith's "A Press of Our Own Kitchen Table: Women of Color Press" (Ahmed 2010; Woolf [1929] 2014; Smith 1989).

Ahmed and other feminists' attention to the spatio-temporal, material dimensions, and socio-cultural dynamics of writing, if extended to digital storytelling practices, is, likewise, effective in reinforcing the immersive and participatory potential of digital storytelling, whether that is through co-authoring a story online, providing feedback on others' stories, presenting an online story to a live and/or remote audience, or connecting with other people and with geographical sites in real time in ways that impact our perception of space, time, and community. Stories are highly contextual and complex, actively engaging their audiences on cognitive, emotional, affective, interpersonal, symbolic, and cultural levels (Ahmed 2010). By providing these young women storytellers with ample opportunity to choose and combine text, images, video, sound, interactive elements, or kinetics—in one story—and, thus, draw on multiple sensory channels at once, we open up diverse ways for them to express themselves and engage their audiences at all of these levels in transformative ways. In the next section, we describe the structure of the workshop and the materials used to foster storytelling.

### 3. Collaborative Storytelling Approach

*3.1. Digital Storytelling Workshop for Young Women*

The "Gathering Stories" workshop employed multiple dimensions of story work to equip and empower young women as storytellers and as co-creators of their life stories, both individually and collectively. The workshop introduced the young participants to guest storytellers from the Rochester community and historic examples, allowing them to understand and themselves practice story work as a transformative tool for women to reflect on and effectively transform their own lives and communities through self-expression, knowledge, and community building. The workshop 'gathered' young women from across the Rochester area and provided them with an educational stipend of $250 upon successful completion of the workshop. This stipend enabled participation by those hardest-hit by the consequences of Rochester city schools' remote learning during the height of the pandemic and subsidized those who might typically be working through the summer. The workshop also 'gathered' together diverse and differently valued kinds of multimodal digital 'writing,' as equally valuable storytelling methods, focusing on storytellers' development of their own voice through the digital storytelling medium of their choice.

*3.2. Workshop Structure*

Our program is organized as an annual one-week, five-day program, running from 9 a.m. until 3 p.m. daily for a group of 10–18 female-identified high school students. In the first iteration of the program in 2021, our 18 participants worked individually (for ideating and writing); in story circles (sharing ideas and writing); and in the full group (instruction, writing exercises and prompts, guest storytellers, sharing ideas). Each day was predicated on a theme (*Connection*, *Engaging*, *Sharing*, *Gathering*, and *Celebrating*) and broken into several segments to maintain a lively pace. We enjoyed a shared lunch every day as well (also provided), where stories were shared in more casual surroundings. Each of the first two days offered a mix of faculty instruction and facilitation of writing and sharing, guest storytellers, longer intensive writing sessions, icebreakers, and exploratory play. Day 3 focused on each participant taking their chosen story (in its draft form) and expanding its aesthetic possibilities through the incorporation of media elements (e.g., photographs, video, illustrations, animation, comics, and podcasts). During Day 4 and the first half of Day 5, participants revisioned and revised their projects, and by the afternoon of the final day, they had completed an edited, shareable story. We also ended each day with the participants documenting their experience in a digital reflective journal that the facilitators had access to, which provided workshop leaders with insight into each individual's progress—even their processes of transformation. On the last afternoon, we hosted a story showcase, inviting family and friends to join us (in person or online) for the culminating presentations and story performances.

*3.3. Workshop Methods*

In order to build connections and foster collaboration between young storytellers, we co-authored ground rules with participants that ranged from asking open-ended and non-judgmental questions to being respectful when others express their ideas and to paying attention to non-verbal cues (modeling protocols under the acronym *CARES: confidentiality*, *acceptance*, *respect*, *experience-centered feedback*, and *support*). We prioritized interpersonal feedback and exchange (via three-person story circles, college student facilitators, and exploratory small-group activities). One of the overarching goals of the workshop was to encourage the group to engage in a practice of *radical listening* to ensure that reciprocity, care, and listening without judgment remained central to our workshop space and interpersonal interactions. Each day's activities were also linked to elements of the StoryCenter model and its "Seven Steps to Creating a Digital Story: *Think It, Feel It, Show It, See It, Hear It, Mix It, Share It*" (Lambert and Hessler 2020). The "Gathering Stories" workshop goals included: documenting experience; addressing vulnerabilities (personal and societal) and empowering positive responses; centering authentic knowledge production; cultivating

interpersonal connections; fostering compassion for self and empathy for others; and facilitating speaking, listening, and writing skills.

Along with our all-female workshop leaders and student assistants with diverse identities, we hosted guest speakers including an award-winning memoirist, a journalist, and a Ghanaian literary scholar (joining us by Zoom). One important goal of the workshop was to practice documenting experiences—both visual and written—and model respect for diverse storytelling methods and kinds of storytellers. This approach allowed participants to leverage their strengths and attempt a new skill. As one young storyteller's journal entry on Day 5 reflects,

> "I really enjoyed seeing everyone's stories/projects. I feel as if it's really inspired me and it's making me strive to be better because everyone's [story] was so unique and very well done... I think I've really learned a lot that will help me with my writing for my upcoming college applications and even for other personal creative projects. Sad to go, but so very excited for the future!"

This is just one example of the thoughtful journal entries our young storytellers wrote at the close of each day, highlighting what worked well and where they struggled. It was refreshing to see the gradual change in their writing and the deepening of the stories they worked on during the week, as well as to see written confirmation of the rapport that we witnessed in our daily interactions.

*3.4. Building Connections with Other Local Teen Storytellers*

Too often, teens keep their most affecting narratives to themselves, offering less authentic stories in the public arena—or remaining silent. Many Gathering Stories participants initially present as quieter and more introverted and report that they have not found a community of writers and storytellers until our program. Finding a comfortable, safe, in-person social connection remains elusive to many youths. Thus, the workshop design and implementation focus on teen participants developing their own stories. Bringing together students from urban, suburban, and rural regions from a cross-section of different socio-economic demographics as well as a diversity of ethnicities and levels of education and writing experience, Gathering Stories provided a separate space a few paces away from familiar high school social scenes and pressures, allowing these high school women to develop affinities as storytellers without these usual concerns.

Emphasizing teens' experiences creates opportunities and presents challenges. In order to create an opportunity for sharing personal experiences, we began the workshop by honoring each participant's initial engagement level and leaving plenty of space for re-orienting and re-visioning as each day, the week, and their storytelling practice and reflection progressed. We asked them to create seed stories to build upon during the workshop and into the future. And we emphasized that although personal writing and self-expression can often be cathartic and the most successful stories are, of course, born out of narrative conflict, there are no expectations that their stories will be those of traumatic life experiences, freeing them to explore a range of narratives, including those that might present as more quotidian, humorous, or low-stakes.

## 4. Results

*4.1. Storytelling Successes*

At the end of the first workshop in 2021, 18 participants completed and shared (in our closing Story Showcase) their final projects, which ranged from a blog exploring cross-cultural experiences of food to an experimental documentary film to a comic following the trajectory of a personal passion to a text-image project demonstrating how love can overcome familial challenges. In daily reflective journal entries and in feedback from participants (and their parents) after the workshop's end, the majority of the young women expressed excitement and appreciation for the wide range of writing activities and storytelling approaches introduced over the week, while differing in their preferences. They indicated significant personal satisfaction in completing their story and sharing it at the end

of the week, as well as some recognition that this was the beginning of a work in progress that they would continue to develop. As one participant posted on her final project web site, "This workshop brought out confidence in my writing that I wasn't aware that I had in me." A parent of another participant later confided that her daughter "hadn't thought her voice mattered," but the workshop changed that and strengthened her resolve to apply for a competitive college program, to which she has now been admitted. In a journal entry for Day 3 of the workshop, titled "Living on a Prayer," one young storyteller wrote:

> "This week went by insanely fast... As per usual, today was pretty dope... There were also a few talks that I enjoyed. The exercises were also quite creative. Overall, I think that I've learned a lot, which is great. Now to pray that my project and site turn out well. Video recording time, woohoo!"

One young storyteller who had been home-schooled through high school titled her project "Diary of an Insecure Filmmaker," using drawn text and family photos to chronicle her development as a visual storyteller; two years later, she is enrolled in a university film program and determined to become a director. Two participants were attending a Title 1 school in Rochester with 96.4% economically disadvantaged students and 92.5% minority students. One of them wrote an illustrated comic about a young wolf who is scorned by the pack for being "weird." This young woman acknowledged in her final journal entry:

> "The storytelling workshop may have been a bit nerve racking due to my terrible anxiety but I still loved the experience. There were many things that I did that were out of my comfort zone but I still found the courage to do it because I wanted to learn from this experience. I also did something that I've been meaning to do myself and because of this workshop... express my art through comic form and I loved it."

Both of the young storytellers from this Title 1 school returned for a "Spooky Halloween Storytelling Event" we held for young women for the next two years, although one of them has yet to graduate from high school. Several young storytellers also expressed their desire to participate in the workshop again next summer. In the next section of this essay, we reflect on the lessons learned from our first offering of the workshop and discuss the ethical implications of teaching multimodal storytelling to young adults.

*4.2. Lessons Learned*

The age demographic of the workshop's participants means they are digital natives, but there also exists a very real digital divide, exacerbated by the recent pandemic. Not all of our participants have access to a computer at home or internet access, making it difficult for some to spend additional time building on the work completed during the workshop or to experiment with the tools introduced during our daytime sessions. While the grant funding allowed us to offer a program that was both free of charge and provided an educational stipend to participants, some interested students were not able to commit for a full week as they needed to work summer jobs. This is a real-world situation we are working to address.

Hosting the workshop on a university campus provided access to various digital platforms to create multimodal stories, expanding writing projects in a way that encourages creative experimentation. The campus, however, is not centrally located, which resulted in transportation challenges for some of our participants, who sometimes were delayed in getting to campus and missed out on some of the activities. The socio-economic diversity of our participants also highlighted differences in writing skills and variances in parental support or caring duties at home. Students who required extra writing help also struggled to complete the homework before the next day's activities, which slowed the progress on their main projects.

Prioritizing experiential dimensions of learning and capturing personal experiences through storytelling meant the participants were encouraged to use their smart devices to

collect information. Thus, keeping all participants engaged in the radical listening process sometimes became an issue when cell phones instead introduced distractions.

## 5. Discussion

Stories are a primary means for individuals and cultures to produce meaning, identify patterns, and transmit these across generations and cultures. From this vantage point, it is no surprise that today we are seeing broad turns toward storytelling, which might be understood as an attempt to ground or stabilize our cultures in the midst of radical change. Yet it is precisely these aptitudes and inclinations to identify and consolidate recognizable, repeatable, and memorable patterns or meta-narratives that can serve simultaneously as a limit and liability to stories in their capacity to calcify, oversimplify, and mislead. The turn to narrative threads like Ronald Reagan and later Donald Trump's "Make America Great Again," for example, is reactive and retrogressive in the desire to return to a wholly idealized, unreal American past, though obviously comforting to many Americans nonetheless. Addressing this conundrum, the philosopher Paul Armstrong usefully situates stories at the crux of our cognitive need to identify patterns while also remaining open to new information and change. He suggests that the complex dynamics of exchanging stories are, in fact, a "two-way, back-and-forth interaction" between the story and reader that might effectively keep "our cognitive processes from congealing into rigid habitual patterns... by holding open their capacity to be reshaped and re-formed" (Armstrong 2020). The legacy of Octavia Butler's fictional *The Parable of the Sower* and the (unfinished) *Parable* trilogy, in particular, provide an instructive fictional precedent in this regard (Butler [1993] 2021). As Armstrong theorizes, stories involve us in the "play of configuration and refiguration," which can "loosen the habitual, ideological hold of any particular set of narrative patterns on our individual and social minds" (Armstrong 2020). Our teen participants are very aware of events, at least at the local and national levels, and their reaction is often to internalize their emotional responses, particularly those that might cause discomfort. A storytelling practice mitigates these tendencies by situating healthy self-expression within a universal milieu that illuminates commonalities and fosters core connections amid differences. While our teen participants are socially aware, they do not necessarily have a strong grasp on the affordances of digital platforms and the implications of sharing personal stories publicly, so demonstrating the power of these tools and practices is a fundamental goal of the workshop.

We fully recognize the importance of creating longer-term consistency and engagement by fostering the community we have built over subsequent months, yet there is the ongoing challenge of how to support the Gathering Stories summer participants, especially those from underserved communities, after the program ends. We offered a few workshops and gatherings throughout the year after the workshop (such as organizing post-workshop themed parties, a logo creation contest, assistance with writing letters of recommendation for college, and an additional storytelling event), but it was admittedly a challenge of scheduling and funding, and some participants aged out of high school and, understandably, moved on in their lives. Thus, one of our lingering questions is how to expand access to multimodal storytelling tools and provide and grow a space for the creation of stories after the program ends. In future iterations of this workshop and related wrap-around storytelling events, we are considering partnerships with the City of Rochester's Libraries and Recreation Centers so that we can hold half-day storytelling events throughout the school year and make these low-risk and high-access for more young storytellers of all genders. These easy-access gatherings, focusing on flash fiction, horror stories, or slam/performance poetry, would complement our more intensive summer workshop and, we hope, open doors to more young people.

We recently launched a web page on the Center for Engaged Storycraft's website to feature our young storytellers and their digital stories annually and archive the workshops, although we are fully cognizant that some of our young storytellers prefer not to have their stories publicly accessible and, therefore, we will ensure consent and keep some stories

password protected. With Chat-GPT and other large-language AI models actively pilfering the web, the question of how and whether to publish these young storytellers' work online at all is an increasingly loaded ethical question. In the short term, we intend to educate our young storytellers about Chat-GPT and other large-language models whose predictive learning methods may impact future writing and our understanding of individual creative work and how it should (or should not) circulate. We will encourage informed choices and explore how generative AI might contribute in the way of brainstorming, audience research, or writing feedback if used with its abilities and blind spots in full view.

## 6. Conclusions

In closing, we acknowledge that the ability to take time out to focus on storytelling is a privilege not available to everyone. Creating a collaborative space for storytelling, in itself, is not sufficient, as intersectionality is about structural power, not just multicultural gatherings or identity. The challenges and tensions of gathering young women, families, and educators who remain extremely stratified along racial, economic, linguistic, and cultural lines and have wholly different experiences of American opportunity are a constant concern whose remedy extends well beyond the workshop space of gathering to broader transformations in the community and in networks of storytelling, power, racial understanding, healing, etc. As educators, we recognize that more work is needed to truly co-realize collaborative spaces where universities can work with communities to address larger structural issues in informed ways and foster multimodal storytelling. To this end, we nonetheless remain committed.

**Author Contributions:** L.S.: Methodology, Writing—original draft, writing review & editing; A.K. and K.v.: Methodology, Writing—original draft, writing review & editing. All authors have read and agreed to the published version of the manuscript.

**Funding:** This research received no external funding.

**Conflicts of Interest:** The authors declare no conflict of interest. The funders had no role in the design of the study; in the collection, analyses, or interpretation of data; in the writing of the manuscript; or in the decision to publish the results.

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
