# Peer review of "Gathering Stories: Creating Spaces for Young Women to Connect and Build Community through Multimodal Storytelling"

_socsci, doi:10.3390/socsci12090487_

Round 1

Reviewer 1 Report

WAYS TO STRENGTHEN:

  1. Increased Connection Between Sections: While each section provides valuable insights, they sometimes feel disjointed. Strengthening the links between the historical context, the workshop's design, and its outcomes could create a more cohesive and engaging narrative. Drawing parallels between the theoretical background of storytelling, the workshop's practical implementation, and the broader societal implications could enhance the text's coherence.
  2. Provide a Balanced Perspective: While the document discusses the successes and positive outcomes of the workshop, there's room for a more nuanced examination of challenges, failures, and lessons learned. Adding this depth would offer a more complete picture, contributing to the credibility of the text. Specific examples or case studies of what went wrong, and how it was (or could be) addressed, would enrich the reader's understanding.
  3. Accessibility and Engagement: The language and tone of the document are quite academic, which might limit its appeal to a broader audience. Injecting more engaging anecdotes, participant testimonials, or visual aids might make the text more relatable and exciting for a wider readership. Crafting the document with more varied audience levels in mind would expand its reach and potential impact.

I thought the writing was good, clear and succinct. I did comment below that the writing could be quite academic. See my comments above. 

Author Response

Increased Connection Between Sections: While each section provides valuable insights, they sometimes feel disjointed. Strengthening the links between the historical context, the workshop's design, and its outcomes could create a more cohesive and engaging narrative. Drawing parallels between the theoretical background of storytelling, the workshop's practical implementation, and the broader societal implications could enhance the text's coherence.

  1. We have strengthened the connections between each section of the article and there is a clear throughline connecting the historical context, the workshop’s design, and its outcomes.
  2. Provide a Balanced Perspective: While the document discusses the successes and positive outcomes of the workshop, there's room for a more nuanced examination of challenges, failures, and lessons learned. Adding this depth would offer a more complete picture, contributing to the credibility of the text. Specific examples or case studies of what went wrong, and how it was (or could be) addressed, would enrich the reader's understanding.
  3. We agree that our discussion of challenges we face and lessons learned in the workshop were cut short at the conclusion of the article. We have expanded the discussion and added specific examples of issues we faced, such as disengagement of some participants, issues with transportation to the campus, lack of parental support, lack of a computer or wifi at home to work on in the evenings, and differences in writing skills among participants. As the article is a theoretical piece reflecting on opportunities digital storytelling provides for engaging high school women, we do not have case studies, per se, but we discuss changes we made in the second workshop offering to address these issues.
  4. Accessibility and Engagement: The language and tone of the document are quite academic, which might limit its appeal to a broader audience. Injecting more engaging anecdotes, participant testimonials, or visual aids might make the text more relatable and exciting for a wider readership. Crafting the document with more varied audience levels in mind would expand its reach and potential impact.
  5. We agree that the article’s introduction, particularly, is academic in tone and style and speaks to the journal’s audience rather than a wider readership as is a goal of this special issue. We have moved the theoretical framework discussion to later in the article and begin with a more accessible and engaging introduction.

Reviewer 2 Report

This is an interesting work that aims to discuss the unrecognized affordances and value of digital storytelling practices for teens living in precarious (neo)colonial life worlds.

Below there are some remarks and suggestions that aim to help the authors strengthen their manuscript prior to publication.

First, the article lacks a thorough and concrete theoretical framework. In its current status this work presents a short theoretical framework (in the Introduction part, pp.1-2) which does not cover the issues discussed here. Also, it is based on a limited number of sources (references). The authors need to expand the theoretical framework and thoroughly revise it based on references regarding, at least, the main issues discussed here, namely digital storytelling and young generations’ understanding of storytelling.

Second, there are some arguments throughout the manuscript that need referencing, i.e., p.5, lines 220-222 (“stories are a primary means for individuals and cultures to produce meaning, identify patterns, consolidate more complex and varied experiences and transmit these across generations, cultures, and places”).

Also, while the procedures, goals, etc. of the workshop conducted are analytically explained, the research questions and method used for the study are not sufficient. The authors are advised to present their method section in a more academic way initiating from the RQs of the study.

Finally, the Conclusions part is a bit poor and needs a thorough revision. What is new here? Why is this study important? Are there any limitations? All these need to be explained.

Author Response

Reviewer 2 - Please find our responses to each of your comments below, in italics.

  1. First, the article lacks a thorough and concrete theoretical framework. In its current status this work presents a short theoretical framework (in the Introduction part, pp.1-2) which does not cover the issues discussed here. Also, it is based on a limited number of sources (references). The authors need to expand the theoretical framework and thoroughly revise it based on references regarding, at least, the main issues discussed here, namely digital storytelling and young generations’ understanding of storytelling.

We have  expanded on and clarified the theoretical framework we are drawing on in our article and workshop. We moved  this theoretical section so that it is later in the article and won’t be an obstacle to readers who are not familiar with this theoretical groundwork, while also clarifying the theoretical assumptions and expertise we drew on in developing this workshop for academic readers interested in this background.

2. Second, there are some arguments throughout the manuscript that need referencing, i.e., p.5, lines 220-222 (“stories are a primary means for individuals and cultures to produce meaning, identify patterns, consolidate more complex and varied experiences and transmit these across generations, cultures, and places”). 

We have added sources to support claims such as the one referenced above by the reviewer. We were trying to strike a balance between an academic article and an interdisciplinary article accessible to wider audiences, as recommended by Reviewer 1, but we have added citations in keeping with the journal and its readers’ expectations and we formatted the manuscript in the  Chicago style, as requested.

3. Also, while the procedures, goals, etc. of the workshop conducted are analytically explained, the research questions and method used for the study are not sufficient. The authors are advised to present their method section in a more academic way initiating from the RQs of the study.

This article is not an empirical research study. It is, instead, a theoretical piece reflecting on opportunities that digital storytelling and strategic workshop methods provide for engaging and empowering high school women. To address this reviewer’s comment while also expanding the accessibility of the article to wider, non-academic audiences, as Reviewer 1 has recommended, we clarified the theoretical assumptions driving the workshop development and its implementation. Expanding our theoretical section and moving it to later in the article allowed us to clarify our methods and theoretical framework, in response to your comment here and in keeping with Reviewer 1 ’s recommendation.

4. Finally, the Conclusions part is a bit poor and needs a thorough revision. What is new here? Why is this study important? Are there any limitations? All these need to be explained.

We expanded on  the conclusion and addressed lessons learned, limitations of the workshop so far and explain why this theoretical approach and methodology are important.  This will also allow us to address the length of the article and ensure it is well beyond the minimum length.

Round 2

Reviewer 2 Report

The authors have revised the manuscript following the Reviewers' suggestions.